# A multimodal electrochemical approach to measure the effect of zinc on vesicular content and exocytosis in a single cell model of ischemia

Ying Wang ⓘ, Chaoyi Gu ⓘ and Andrew G. Ewing ⓘ

Department of Chemistry and Molecular Biology, University of Gothenburg, 41296 Gothenburg, Sweden

**Key words:**
Fraction of release; ischemia and zinc; ischemia regulated exocytosis; oxygen–glucose deprivation and reperfusion; single-cell amperometry; vesicular content

**Author for correspondence:**
Andrew G. Ewing, E-mail: andrewe@chem.gu.se

## Abstract

Zinc ion is essential for normal brain function that modulates synaptic activity and neuronal plasticity and it is associated with memory formation. Zinc is considered to be a contributing factor to the pathogenesis of ischemia, but the association between zinc and ischemia on vesicular exocytosis is unclear. In this study, we used a combination of chemical analysis methods and a cell model of ischemia/reperfusion to investigate exocytotic release and vesicular content, as well as the effect of zinc alteration on vesicular exocytosis. Oxygen–glucose deprivation and reperfusion (OGDR) was used as an *in vitro* model of ischemia in a model cell line. Exocytotic release and vesicular storage of catecholamine content were increased following OGDR, resulting in a higher fraction of release during exocytosis. However, zinc eliminated these increases following OGDR and the fraction of release remained unchanged. Understanding the consequences of zinc accumulation on vesicular exocytosis at the early stage of OGDR should aid in the development of therapeutic strategies to reduce ischemic brain injury. As the fraction released has been suggested to be related to presynaptic plasticity, insights are gained towards deciphering ischemia related memory impairment.

## Introduction

Stroke, a consequence of ischemia, is one of the leading causes of death and disability worldwide. Oxygen and glucose are needed constantly to maintain normal brain function. The lack of oxygen and glucose, even for a short duration, results in irreversible brain injury. Neurons communicate with each other through synaptic transmission and failure of synaptic transmission leads to neuronal dysfunction (Puig *et al.,* 2018). Therefore, to understand how stroke affects the function of the central nervous system, it is important to understand how stroke influences the exocytotic machinery. Exocytosis is typically triggered by an increase in intracellular free $Ca^{2+}$, which allows synaptic vesicles to fuse with the plasma membrane forming fusion pores and releasing vesicular content into the extracellular space. Partial release of transmitters has been recently proposed as the main mechanism of release during regular exocytosis, and is now thought to represent the vast majority of what were originally thought to be full exocytotic events (Omiatek *et al.,* 2010; Li *et al.,* 2015, 2016; Ren *et al.,* 2016; Phan *et al.,* 2017; Ranjbari *et al.,* 2019; Wang and Ewing, 2021). However, it is not known if ischemia has an effect on vesicular storage of neurochemicals or the fraction of molecules released from vesicles during exocytosis.

Zinc is the second most abundant transition metal ion and plays a crucial role in maintaining normal functions of the brain, including brain development and learning and memory (Wang *et al.,* 2020). Zinc is widely distributed in the brain and high concentrations of zinc are found in many synaptic vesicles, suggesting that zinc plays a role in neuronal synaptic transmission (Shuttleworth and Weiss, 2011). Zinc exhibits biphasic effects during ischemia, exerting either neuroprotective or neurotoxic effects. Subacute administration of zinc exerts a neuroprotective effect by decreasing cell death and preventing the loss of memory after transient hypoxia–ischemia (Blanco-Alvarez *et al.,* 2015). However, extracellular accumulation of zinc could also induce apoptosis and neuronal cell death during ischemic injury. Previous studies from our group have shown that zinc regulates the dynamics of exocytosis and vesicular content, by slowing down the exocytotic process and reducing a larger fraction from vesicular storage (Ren *et al.,* 2017, 2019). However, the mechanism by which zinc contributes to the exocytotic release and vesicular content of neurochemical messengers under ischemia is not fully known and there are still many open questions about how zinc affects exocytosis.

In this article, we examined the effect of zinc on exocytosis in PC12 cells as a model during oxygen-glucose deprivation and reperfusion (OGDR), an *in vitro* model for ischemia-like conditions. Many drug treatments and effective protective protocols exhibit multiple protective effects when added at the start of reperfusion. Thus, it is important to assess the molecular machinery of vesicular exocytosis immediately at the start of reperfusion. To study the early

**CAMBRIDGE**
UNIVERSITY PRESS

effects of reperfusion, by restoration of oxygen and glucose, exocytotic release and vesicular content were measured within 1 h after reperfusion began. Single-cell amperometry (SCA) was used to measure exocytosis of catecholamine and intracellular vesicle impact electrochemical cytometry (IVIEC) was used to measure vesicular storage of catecholamine. The numbers of catecholamine molecules released during exocytosis and stored in vesicles were observed to increase during OGDR. Zinc treatment, however, eliminated both increases induced by OGDR, suggesting zinc to be protective at the early stage of ischemia.

## Results

### A model of ischemia enhances quantal release and total vesicular content of catecholamine

To investigate exocytosis of catecholamine during ischemia, oxygen–glucose deprivation (OGD) was performed for 4 h to simulate an ischemia-like condition *in vitro* in PC12 cells. Exocytosis of catecholamine was measured right after OGD, at the start of reperfusion, where oxygen and glucose were restored. A disk-shaped electrode was placed in close proximity to the surface of the cell and SCA was used to quantify exocytosis. The cell was stimulated with 100 mM $K^+$ solution to evoke catecholamine secretion. Amperometric spikes were generated due to the oxidation of catecholamine at the surface of the electrode and each spike corresponds to exocytosis from single vesicles. The number of catecholamine molecules released during exocytosis can be quantified according to Faraday's law ($Q = nNF$), where $Q$ is the total charge transferred, $n$ is the number of exchanged electrons in the reaction, $N$ is the number of moles of catecholamine released from the vesicle and $F$ is the Faraday constant, 96,485 C/mol.

Example amperometric traces show the differences between exocytotic release of catecholamine from a control cell and a cell with OGDR treatment (Fig. 1a,b). The exocytotic events were more frequent in OGDR-treated cells than control cells with a 5 s stimulation duration of 100 mM $K^+$. By comparing the average number of release events, the number of exocytotic events per cell with OGDR treatment was significantly increased (Fig. 1e, $p = 0.013$). The average numbers of events are $15 \pm 2$ events in control cells *versus* $36 \pm 7$ events in OGDR-treated cells. In addition, an elevated cytoplasmic $Ca^{2+}$ level was observed in OGDR-treated cells (Fig. 2), where the elevated $Ca^{2+}$ level likely results in more vesicles fusing with the cell membrane. This is consistent with a larger number of exocytotic events with OGDR-treated cells compared to the control cells. Notably, OGDR significantly enhanced the number of released catecholamine molecules per exocytotic event with an average number of $208,000 \pm 26,000$ molecules, compared to $105,000 \pm 6,000$ molecules from control cells (Fig. 1f, $p < 0.0001$). Thus, OGDR treatment affects exocytosis of catecholamine by increasing the amount of catecholamine molecules released and the number of exocytotic events.

The characteristic shape of each amperometric spike can be used to determine dynamic information about exocytotic release (Fig. 3a). A significant increase in maximum peak current, $i_{max}$, was observed after OGDR treatment (Fig. 3b, $p = 0.0008$), indicating OGDR might induce more catecholamine flux through the open fusion pore. The half width of each spike, $t_{half}$, which represents the duration of the exocytotic event or the duration of the open exocytotic pore, did not change with OGDR treatment (Fig. 3c, $p = 0.91$). OGDR treatment significantly increased the peak rise time, $t_{rise}$, but had no effect on both $t_{half}$ and the peak

decay or fall time, $t_{fall}$ (Fig. 3c–e, $p = 0.0035$, $p = 0.91$, and $p = 0.50$, respectively). Hence, we conclude that the result of OGDR treatment does not affect the overall duration of exocytotic events and the time for the closing process of the fusion pore, but that the opening of the fusion pore is slower and longer resulting in a greater flux of vesicular catecholamine molecules through the fusion pore while it is open.

### Zinc suppresses the OGDR-induced increase in exocytotic release

Accumulation of extracellular zinc has been found in both focal and global ischemia models (Wei *et al.,* 2004; Frederickson *et al.,* 2006). Extracellular zinc can modulate synaptic transmission by interacting with ion channels, transporters, and receptors (Smart *et al.,* 2004; Mott *et al.,* 2008; Veran *et al.,* 2012; Grauert *et al.,* 2014). In previous studies, we showed that 100 μM zinc treatment decreases vesicular catecholamine content and slows down exocytotic release, but does not significantly change the number of molecules released (Ren *et al.,* 2017, 2019). Therefore, we employed the same concentration of zinc (100 μM) to further understand how it affects exocytosis under conditions to mimic ischemia in PC12 cells. Fig. 1c,d show examples of amperometric traces for zinc treatment alone and the combination of zinc and OGDR treatment (zinc + OGDR). An increased frequency of exocytotic events was observed in zinc + OGDR-treated cells compared to the only zinc-treated cells. The number of exocytotic events with the combination of zinc + OGDR treatment was significantly higher with an average of $18 \pm 3$ events per cell, compared to zinc treatment alone with an average of $8 \pm 1$ events per cell (Fig. 1e, $p = 0.026$). However, a lower amount of $Ca^{2+}$ influx was observed with zinc + OGDR-treated cells after stimulation for exocytosis compared to zinc alone (Fig. 2). OGDR treatment increased the number of released catecholamine molecules; however, with zinc treatment, the number of released catecholamine molecules was no longer increased with OGDR. The number of released catecholamine molecules during exocytosis with zinc treatment alone was slightly lower, but not significantly different from the combination of zinc + OGDR treatment (Fig. 1f, $p = 0.26$). On average, there are $103,000 \pm 19,000$ molecules released during exocytosis with zinc treatment alone and $116,000 \pm 11,000$ molecules released following the combination of zinc and OGDR treatment. In addition, there were slight, but insignificant increases and decreases in the maximum peak current, $i_{max}$, and spike fall time, $t_{fall}$, respectively, for zinc + OGDR-treated cells (Fig. 3b,e, $p = 0.13$ and $p = 0.17$, respectively). There was no significant difference in the spike rise time (Fig. 3d, $p = 0.85$) between zinc alone and zinc plus OGDR treatment, indicating the time of the opening process of the fusion pore was not affected. However, zinc plus OGDR treatment significantly decreased the duration of exocytotic event, $t_{half}$, suggesting a less stable fusion pore was formed during zinc plus OGDR treatment compared to cells treated with only zinc (Fig. 3c, $p = 0.039$). Thus, zinc treatment along with OGDR does not affect the number of catecholamine molecules released, but the duration of exocytotic pore tends to be shorter and there is an increased number of exocytotic events.

### Vesicular storage of catecholamine with/without zinc treatment during OGDR

To better understand the effect of OGDR and zinc treatment on exocytosis, we quantified the total catecholamine content that is stored in single vesicles. IVIEC was introduced by our group to

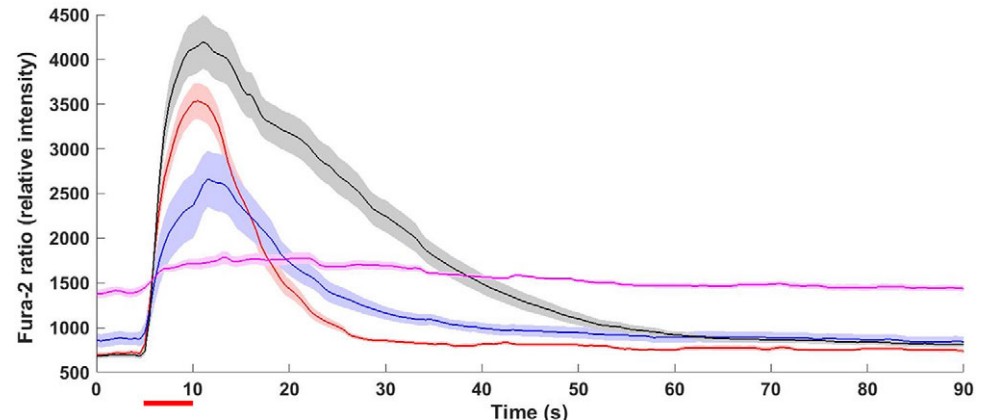

**Fig. 1.** Effect of oxygen–glucose deprivation and reperfusion (OGDR) with/without zinc treatment on exocytotic release. Representative amperometric traces from (*a*) A control cell, (*b*) A cell with OGDR, (*c*) A cell treated with zinc (100 µM) and (*d*) A cell treated with the combination of zinc and OGDR (zinc + OGDR). (*e*) Number of exocytotic events per cell among different conditions. (*f*) Comparison of the number of molecules released per exocytotic events. $n > 16$ cells. *$p < 0.05$, **$p < 0.01$, and ****$p < 0.0001$ (Mann–Whitney rank-sum test).

**Fig. 2.** Intracellular $Ca^{2+}$ levels before, during and after 5-s stimulation with 100 mM $K^+$ stimulation solution. Average $Ca^{2+}$ response from control cells (red), 100 µM zinc treated cells (blue), oxygen–glucose deprivation and reperfusion (OGDR)-treated cells (black), and cells treated with the combination of zinc and OGDR (pink). $n > 30$ cells were imaged for each condition and the shaded areas around the lines represent SEM. The red bar underneath the graph indicates the stimulation period.

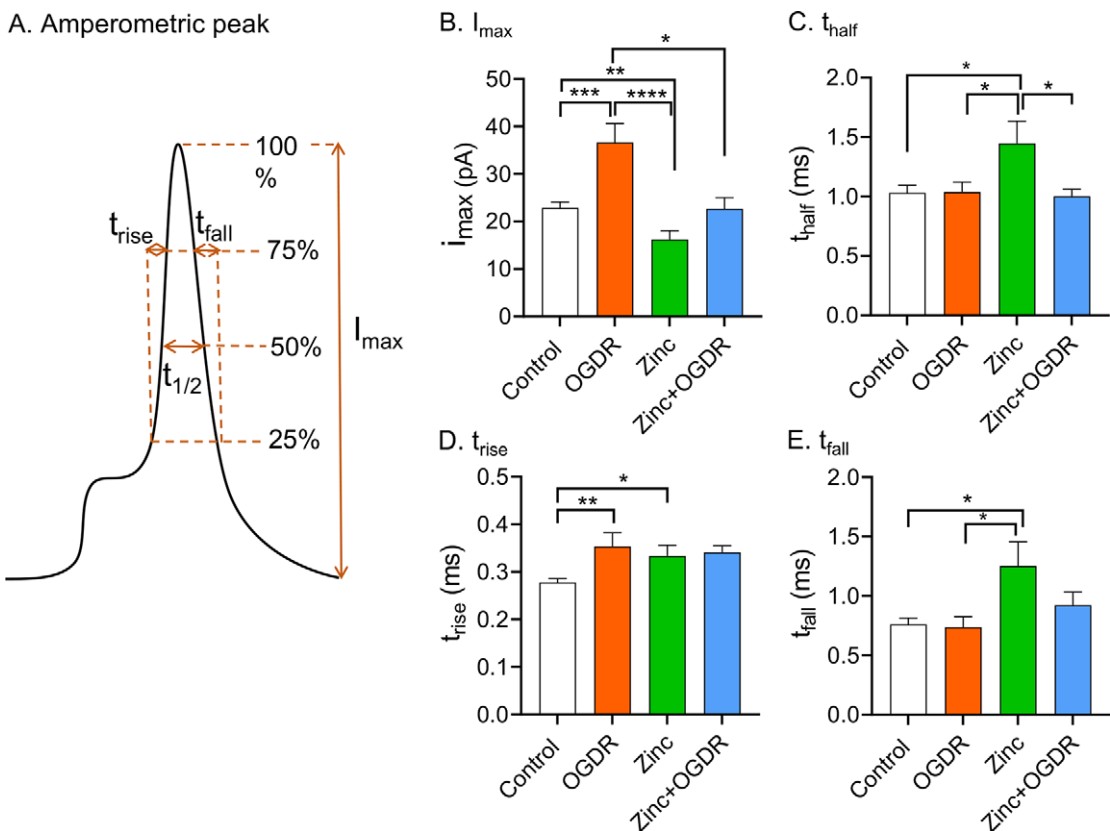

**Fig. 3.** Amperometric peak analysis. (*a*) Schematic of amperometric spike with different parameters. Comparisons of (*b*) Peak current, (*c*) Half peak width, (*d*) Rise time, (*e*) Fall time from single-cell amperometry from control cells, cells treated with zinc (100 μM), cells with oxygen–glucose deprivation and reperfusion (OGDR), and cells treated with the combination of zinc and OGDR (zinc + OGDR). The rise time of each spike, $t_{rise}$, is defined as the time from 25 to 75% of the peak height on the rising part of the peak and corresponds to the opening time of the fusion pore. The fall time of each spike, $t_{fall}$, is defined as the time from 75 to 25% of the peak height on the falling part of the peak and corresponds to the closing time of the fusion pore. $n > 16$ cells. *$p < 0.05$, ** $p < 0.01$, ***$p < 0.001$ and ****$p < 0.0001$ (Mann–Whitney rank-sum test).

directly quantify vesicular content inside a single cell (Li *et al.*, 2015). In IVIEC, a flame-etched carbon fibre nanotip electrode is used to penetrate the cell membrane of a live cell. The intracellular vesicles then adsorb on the electrode surface and rupture by electroporation to release their contents. Fig. 4*a*,*b* show examples of amperometric traces of vesicular content in a control cell and an OGDR-treated cell. The vesicular content of catecholamine with OGDR treatment was significantly higher with an average of 299,000 ± 37,000 molecules, compared to the control cells with an average of 176,000 ± 10,000 molecules (Fig. 4*e*, $p = 0.0003$). Thus, OGDR treatment induces an increase of catecholamine storage in vesicles.

Fig. 4*c*,*d* show examples of amperometric traces of vesicular content in a zinc-treated cell and a cell treated with both zinc and OGDR. The average number of catecholamine molecules stored in vesicles was significantly higher in zinc + OGDR-treated cells with an average of 205,000 ± 15,000 molecules, compared to the only zinc-treated cells, 126,000 ± 5,000 molecules (Fig. 4*e*, $p = 0.0002$). However, the storage of catecholamine molecules in zinc + OGDR-treated cells was significantly lower than OGDR-treated cells ($p = 0.026$), suggesting that zinc suppresses the OGDR-induced increase of catecholamine storage in vesicles.

The fraction of vesicular release is determined as the total molecules being released during exocytosis divided by the total molecules being stored in vesicles. The fraction of vesicular release was 60% in control cells, 70% in OGDR-treated cells, 82% in zinc treated cells and 57% in combined zinc and OGDR-treated cells

(Fig. 4*f*). This suggests that cells release a higher fraction of their vesicular catecholamine content in response to zinc or OGDR treatment, while combined treatment of zinc and OGDR does not affect the fraction of release.

## Discussion

We investigated the effect of OGDR, an ischemia-like condition, with and without zinc on exocytosis and vesicular storage of catecholamines in PC12 cells. To study the early effects of reperfusion, by restoration of oxygen and glucose, exocytotic release and vesicular content were measured within 1 h after reperfusion began. OGDR treatment enhances the quantal catecholamine release and vesicular content, while the OGDR-evoked increases in exocytotic release and vesicular storage are suppressed by zinc.

Energy failure in OGD-induced ischemia leads to an increase in intracellular $Ca^{2+}$ level by enhancing the entry of extracellular $Ca^{2+}$. Previous studies have shown that hypoxia can induce inhibition of oxygen sensitive $K^+$ channels, leading to membrane depolarization and $Ca^{2+}$ influx through voltage-gated $Ca^{2+}$ channels (Conforti *et al.*, 2000). An increased expression of mRNA for oxygen sensitive $K^+$ channels has been observed under chronic hypoxia, which increases the inhibition of the $K^+$ current (Conforti and Millhorn, 1997). An elevated cytoplasmic $Ca^{2+}$ level via increased amounts of $Ca^{2+}$ influx is observed in OGDR-treated cells, where a persistent influx of $Ca^{2+}$ is observed with OGDR treatment and it takes a longer time to bring $Ca^{2+}$ back to baseline level. Excess $Ca^{2+}$ can activate

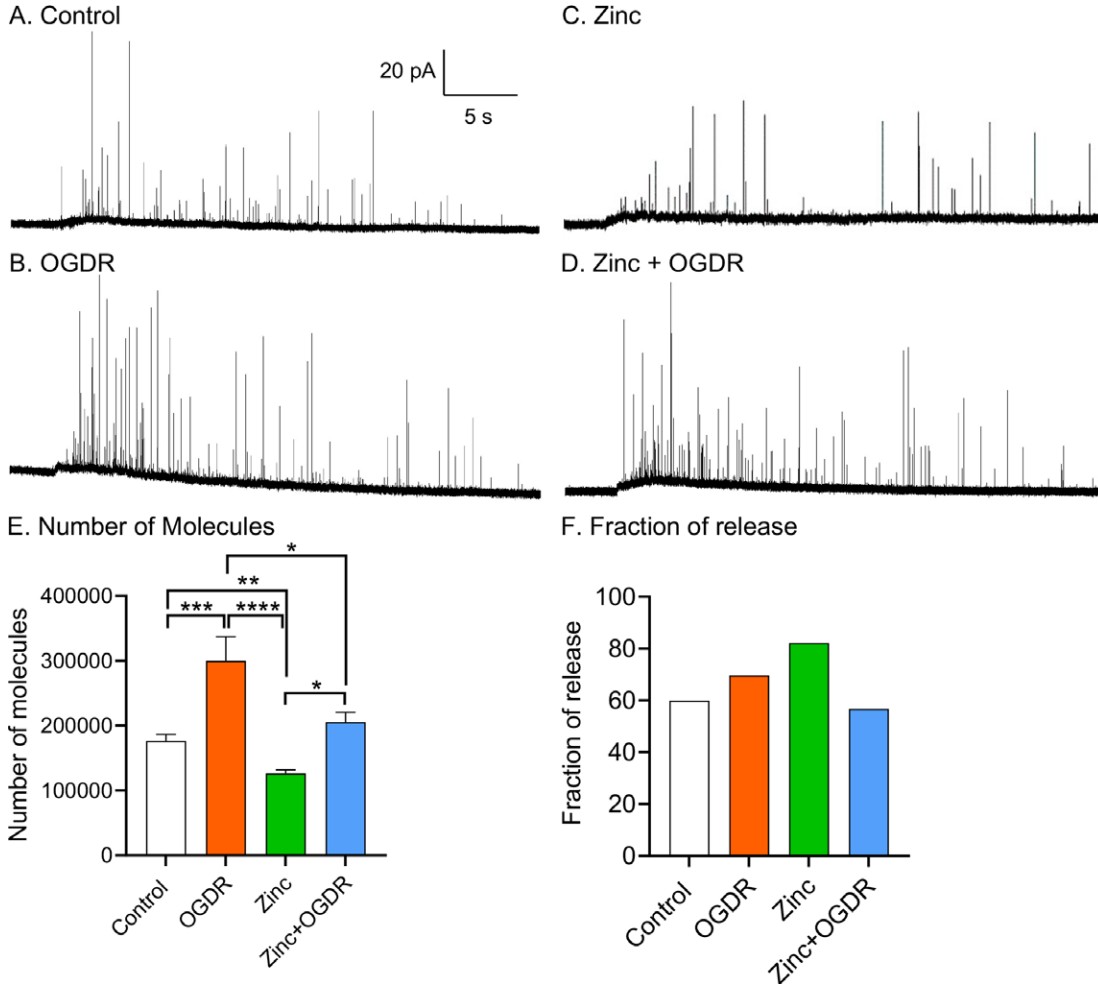

**Fig. 4.** Effect of oxygen–glucose deprivation and reperfusion (OGDR) with/without zinc treatment on vesicular storage and fraction of release. Representative amperometric traces of vesicular content from (*a*) A control cell, (*b*) A cell with OGDR treatment, (*c*) A cell treated with zinc (100 μM) and (*d*) A cell treated with the combination of zinc + OGDR. (*e*) Number of molecules stored per vesicle and (*f*) fraction of release calculated for each condition. *n* > 16 cells. *$p < 0.05$, **$p < 0.01$, ***$p < 0.001$ and ****$p < 0.0001$ (Mann–Whitney rank-sum test).

calcineurin and then induce activation of cofilin. Cofilin is known as a ubiquitous actin-binding protein that modulates actin cytoskeleton dynamics. High activation of cofilin via dephosphorylation has been found during ischemic conditions, where 4 h of OGD completely activates cofilin (Madineni *et al.,* 2016). Activation of cofilin can severely affect actin dynamics, resulting in actin cytoskeleton disruption by depolymerization (Maiti *et al.,* 2015). It has been reported that latrunculin A, a reagent to depolymerize actin filaments and inhibit actin cross-linking, lengthens the time of exocytotic events, enhances the quantal transmitter release and induces a higher number of events (Trouillon and Ewing, 2014). This thus explains the OGDR-induced increases in the fusion pore opening time and the number of molecules released, as well as the higher number of events during exocytosis.

Vesicular catecholamine content is increased with OGDR treatment and this results in a higher fraction of release (70%). This higher fractional release of catecholamine suggests that catecholamine release during ischemia is also via a partial release mechanism as proposed for exocytosis in these cells. A previous study showed that loading of neurochemicals into vesicles requires energy provided by ATP (Larsson *et al.,* 2019). During OGD, oxygen and glucose are

removed, resulting in quick depletion of ATP and therefore, it is expected that OGD will induce a lower or no change of vesicular content. In contrast, an increase of vesicular content of catecholamine after OGD is observed. Reintroduction of oxygen and glucose allows regeneration of ATP. However, a short period of time for reperfusion might only have a minor impact on vesicle loading. Reduction of oxygen can induce an increase of mRNA expression for tyrosine hydroxylase (TH) (Czyzyk-Krzeska *et al.,* 1994), which is the rate-limiting enzyme of the catecholamine synthesis. An increased mRNA expression for TH would then enhance catecholamine levels so that greater amount of catecholamine can be loaded in each vesicle. The vesicular content of catecholamine increased with OGDR, more loaded vesicular content might also result in more release of catecholamine during exocytosis. Thus, the enhanced exocytotic release of catecholamine with OGDR treatment is probably caused by the increased vesicular storage of catecholamine, but perhaps it is partially due to the disruption of the actin cytoskeleton changing the dynamics of the fusion pore.

Zinc treatment decreases vesicular catecholamine content but the amount of exocytotic release is unchanged. Our results are in agreement with previous studies (Ren *et al.,* 2017, 2019), and also

demonstrate that the number of exocytotic events is decreased in zinc-treated cells in comparison to control cells. Additionally, the decreased vesicular content and no change in the number of catecholamine released during exocytosis show that zinc treatment evokes a higher fraction of release (82 *vs.* 60% of control cells) during exocytosis. Extracellular zinc can modulate synaptic transmission by interacting with various ion channels, transporters, and receptors (Vergnano *et al.,* 2014). Zinc accesses the intracellular space through a variety of routes, including $Ca^{2+}$-permeable *N*-methyl-D-aspartate (NMDA) channels and voltage-gated $Ca^{2+}$ channels. Zinc can bind to NMDA receptors and prevent or only allow a small amount of $Ca^{2+}$ from entering the cell. PC12 cells contain $Ca^{2+}$-permeable NMDA channels and voltage-gated $Ca^{2+}$ channels, which are sensitive to zinc inhibition (Legendre and Westbrook, 1990; Casado *et al.,* 1996; Liu *et al.,* 1996; Kerchner *et al.,* 2000). A lower amount of $Ca^{2+}$ influx was observed with zinc-treated cells after stimulation for exocytosis, likely causing less vesicles to fuse with the cell membrane and thus, less number of exocytotic events was observed during exocytosis.

Zinc is highly concentrated in synaptic vesicles in glutamate neurons and also co-localised with GABA and glycine in synapses (Blakemore and Trombley, 2017). During ischemia, more vesicles fuse with the cell membrane to release their content leading to a significant extracellular zinc accumulation, which can cause neuronal death (Frederickson *et al.,* 2006; Kitamura *et al.,* 2006). In this study, exogenous zinc was added to mimic zinc accumulation during ischemia to study the effect of zinc and ischemia on exocytosis and vesicular storage of neurochemicals. While OGDR increases the number of catecholamine molecules released during exocytosis, that increase is suppressed when zinc is added. In addition, elevated basal $Ca^{2+}$ level and only a small rise in cytosolic $Ca^{2+}$ after stimulation were observed with combined zinc and OGDR. Cytosolic $Ca^{2+}$ level is regulated by mitochondria and their calcium loading capacity. Accumulated zinc during ischemia can enter mitochondria via a mitochondrial calcium uniporter that induces mitochondrial dysfunction to decrease calcium loading capacity and promotes cytosolic $Ca^{2+}$ overload (Iijima *et al.,* 2008; Tanaka *et al.,* 2008; Ji *et al.,* 2020). Thus, the elevated basal cytosolic $Ca^{2+}$ level is likely due to the release of $Ca^{2+}$ from mitochondria. Treatment of zinc decreases intracellular $Ca^{2+}$ response to the stimulation, while the combination of zinc and OGDR treatment even exacerbates this effect. This inhibitory effect indicates the combination of zinc and OGDR suppresses intracellular $Ca^{2+}$ elevation in stimulated exocytosis. The OGDR-induced increase in cytosolic $Ca^{2+}$ concentration is likely inhibited by zinc, where zinc serves as a channel blocker for NMDA channels and voltage-gated $Ca^{2+}$ channels, attenuating the influx of $Ca^{2+}$. Excessive release of glutamate during ischemia causes overactivation of NMDA receptors, leading to a massive $Ca^{2+}$ influx and excitotoxicity (Dong *et al.,* 2009). Previous studies on zinc accumulation on ischemic neuronal injury have shown that a low concentration (under 100 μM) of zinc is neuroprotective, as both glutamate-induced $Ca^{2+}$ influx and neuronal death are inhibited in cultured neurons (Kitamura *et al.,* 2006). However, a high concentration (over 150 μM) of zinc is toxic and causes neuronal death. Thus, a low concentration of extracellular zinc accumulation may be protective, by binding to NMDA receptors to prevent the consequences of NMDA receptor overactivation as well as by limiting ischemia-induced exocytosis.

Vesicular storage of catecholamine molecules with zinc plus OGDR treatment is higher when compared to zinc treatment alone, but lower when compared to the OGDR treatment, suggesting that zinc also suppresses the OGDR-induced increase in vesicular storage of catecholamines. OGDR treatment increases the number of exocytotically released molecules and vesicular catecholamine content, whereas zinc treatment alone decreases the vesicular content and results in no change in exocytotic release *versus* control. These opposite effects possibly offset each other and lead to no change in the fraction of release (57 *vs.* 60% in control) for zinc plus OGDR treatment. Interestingly, these data suggest that extracellular zinc accumulation can be induced by treating with a low and not toxic dose of zinc at the early stage of ischemia. This will attenuate the ischemia-induced increases in both transmitter release during exocytosis and vesicular storage, thus maintaining a constant fraction of release during partial exocytosis.

## Conclusions

Our results demonstrate that OGDR enhances the amount of catecholamine released during exocytosis and the vesicular storage of catecholamine, resulting in a higher fraction of release. However, treatment with zinc attenuates the increased amount of catecholamine released and vesicular content induced by OGDR, so the fraction of release remains unchanged. Partial vesicular release has been suggested to be the primary mechanism during regular exocytosis and this mode of release was also observed under ischemic conditions. We suggest that low levels of extracellular zinc accumulation at the early stage of ischemia help to reduce ischemic dysfunction that is caused by excessive catecholamine release. To reduce transmitter release and thus, hyperexcitability during ischemic conditions, zinc would ultimately contribute to decreased excitability in the post-ischemic conditions and recovery. In addition, the consequences of extracellular zinc accumulation on exocytosis and vesicular storage during ischemia might also help to understand ischemia related changes in synaptic plasticity and memory impairment.

## Materials and methods

### Chemicals

All components of the isotonic solution (150 mM NaCl, 5 mM KCl, 1.2 mM $MgCl_2$, 2 mM $CaCl_2$, 5 mM glucose and 10 mM HEPES), the $K^+$ stimulation solution (55 mM NaCl, 100 mM KCl, 1.2 mM $MgCl_2$, 2 mM $CaCl_2$, 5 mM glucose and 10 mM HEPES), and the phosphate-buffered saline (PBS) solution (3.0 mM KCl, 10.0 mM $NaH_2PO_4$, 2.0 mM $Na_2SO_4$, 1.2 mM $MgCl_2$, 131.25 mM NaCl, and 1.2 mM $CaCl_2$) were purchased from Sigma-Aldrich (Stockholm, Sweden). The pH was adjusted to 7.4 using concentrated HCl or NaOH. All aqueous solutions were prepared using 18 MΩ.cm Milli-Q water from Purelab Classic purification (ELGA, Sweden) and filtered using the vacuum filtration system with a membrane pore size of 0.45 μm (VWR, Sweden).

A 10 mM stock solution of dopamine (Sigma-Aldrich, Sweden) was prepared in 0.1 M $HClO_4$ (Sigma-Aldrich, Sweden) and stored at −20 °C. A 100 μM dopamine solution was prepared daily in PBS buffer for testing electrodes. 100 μM $Zn^{2+}$ was prepared daily in RPMI 1640 medium (Sigma-Aldrich, Sweden). This concentration of $Zn^{2+}$ was selected based on literature (Ren *et al.,* 2017, 2019).

### Cell culture

PC12 cells, a gift from Lloyd Greene at the Columbia University, were maintained in RPMI 1640 medium supplemented with 10%

donor horse serum (Sigma-Aldrich, Sweden) and 5% fetal bovine serum (Sigma-Aldrich, Sweden) in a 100% humidified incubator with 7% $CO_2$ at 37 °C. The cells were grown on collagen type IV coated cell culture flasks (Corning BioCoat, Fisher Scientific, Sweden) and sub-cultured every 7–9 days. The medium was replaced every 2 days. For experiments, cells were sparsely-seeded on 60 mm collagen type IV coated cell culture dishes (Corning BioCoat, Fisher Scientific, Sweden) for 4–5 days prior to the experiments. Right before the experiment, the medium was removed and the cells were washed three times with isotonic solution and kept in isotonic solution for the experiments. The experiments were performed at 37 °C.

### Oxygen–glucose deprivation and reperfusion (OGDR)

Prior to OGD, the culture medium was removed, and the cells were washed twice with and bathed in glucose-free medium (Gibco, Fisher Scientific, Sweden) supplemented with 2% donor horse serum and 1% fetal bovine serum. OGD was induced by placing the cells in a modular incubator chamber (MIC-101, Billups-Rothenberg Inc., San Diego, CA). The chamber was checked for leaks before each experiment and was flushed with a mixture of 5% $CO_2$ and 95% $N_2$ at 20L min$^{-1}$ flow rate controlled by a flow metre (DFM-3002, Billups-Rothenberg Inc., San Diego, CA) for 20 min to completely remove oxygen. The chamber was then maintained at 37 °C for 4 h. After OGD, cells were replenished with isotonic solution that contained glucose and re-oxygenated to simulate reperfusion. Meanwhile, control cells were cultured with regular culture medium and kept under normal oxygen condition.

### Fabrication of carbon fibre electrodes and electrochemical measurements

Carbon fibre electrodes were fabricated by aspirating a 5 µm diameter carbon fibre into a glass capillary (O.D.: 1.2 mm, I.D.: 0. 69 mm, 10 cm length, Sutter Instrument Co., Novato, CA) and pulling it into two separate electrodes with a vertical micropipette puller (model PE-21, Narishige, Inc., Japan). For disk electrodes, the extended carbon fibre was cut with a scalpel to the edge of the glass and the fibre/glass interface was sealed with epoxy (G A Lindberg ChemTech AB, Sweden). Electrodes were cured at 100°C overnight and were then bevelled with a commercial beveller (EG-400, Narishige Inc., London, UK) at a 45°C angle. For nanotip electrodes, the extended carbon fibre was cut to a length of 100–150 µm and then flame-etched with a butane gas burner (Clas Ohlson, Sweden) to obtain a thin needle shape with 50–100 nm tip diameter and 30–100 µm length. The electrodes were then sealed with epoxy. Excess epoxy was rinsed with acetone and the electrodes were subsequently cured at 100 °C overnight. Both disk and nanotip electrodes were tested with cyclic voltammetry (−0.2 V to 0.8 V *vs.* Ag/AgCl, 100 mV s$^{-1}$) in 100 µM dopamine solution. Electrodes showing good response to dopamine and stable steady-state currents were used for the experiments.

Amperometry was used to detect exocytotic catecholamine release and vesicular content. Electrochemical measurement was performed at single cells on an inverted microscope (IX71, Olympus). The electrodes were held at a constant potential of 700 mV *versus* an Ag/AgCl reference electrode with an Axopatch 200B potentiostat (Molecular Devices, Sunnyvale, CA). The output was filtered at 2 kHz and digitised at 5 kHz. Exocytosis was stimulated with $K^+$ stimulation solution using a microinjection system

(Picospritzer II, General Valve Corporation, Fairfield, NJ) and each injection was triggered for 5 s with a 20-psi injection pulse.

### $Ca^{2+}$ imaging

PC12 cells were seeded on 60 mm collagen type IV coated dishes for $Ca^{2+}$ imaging experiments. Fura-2 was used as the fluorescence dye for $Ca^{2+}$ imaging. The Fura-2 stock solution was prepared by mixing one vial of Fura-2 (50 µg, Invitrogen, Fisher Scientific, Sweden) with 100 µl dimethyl sulfoxide (DMSO). Before Fura-2 incubation, the cell medium was removed and the cells were washed two times with isotonic solution. Fura-2 incubation was achieved by adding 4 µl of the Fura-2 stock solution into 5 ml isotonic solution, resulting in a final concentration of 0.4 µM. The cells were incubated for 20 min at 37 °C. Afterwards, the cells were washed three times with isotonic solution and maintained in the isotonic solution for the experiments.

Fluorescence imaging was performed using an inverted microscope (IX71, Olympus), which had an MT-20 illumination system and employed a 150 W Xenon arc lamp as the light source. Imaging of the intracellular Fura-2 level was done ratiometrically at the wavelengths of 340 and 380 nm for 90 s. Images were acquired with Cell-R software (Olympus) and an ORCA-ER camera (Hamamatsu, Japan). After a baseline of 5 s, exocytosis was induced by a 5-s stimulation with the $K^+$ stimulation solution, the same condition as the electrochemical measurement. Background subtraction and the analysis of regions of interest were performed in the Cell-R software and the intracellular $Ca^{2+}$ changes under different experimental conditions were analysed using Matlab (The MathWorks Inc.).

### Data analysis and statistics

All data were converted in Matlab (The MathWorks Inc.) and analysed with an Igor Pro 6.7 script (from David Sulzer, Columbia University). A 1-kHz (binomial sm.) filter was applied to all amperometric traces and the threshold for peak detection was five times the standard deviation of the noise. Amperometric traces were inspected manually after peak selection to avoid false positives. All statistics were performed in GraphPad Prism 8 (GraphPad Software Inc., San Diego, CA) with unpaired Mann–Whitney rank sum test. Statistical significance was designated at $p < 0.05$ and all data are presented as mean ± SEM.

**Acknowledgements.** The European Research Council (ERC Advanced Grant Project No. 787534 NanoBioNext), Knut and Alice Wallenberg Foundation, and the Swedish Research Council (VR Grant No. 2017-04366) are acknowledged for financial support.

**Conflict of interest.** The authors declare no conflicts of interest.

**Author contributions** A.G.E. and Y.W. conceived the idea and designed the study. Y.W. performed SCA and IVIEC experiments, analysed the data and wrote the manuscript. C.G. performed IVIEC and calcium imaging experiments. A.G.E., Y.W. and C.G. were involved in data discussion, data interpretation, reviewing and editing the manuscript and read and approved the final version of the manuscript.

**Data availability statement.** The data that support the finding of this study are available from the corresponding author upon reasonable request.

**Open Peer Review.** To view the open peer review materials for this article, please visit http://doi.org/10.1017/qrd.2021.10.

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
