## [Reviewer Report]

*Comments to Author*: The paper of Ying Wang et al.is devoted to clarification of the effect of zinc on exocytosis

and vesicular storage of catecholamine observed after ischemia modeled by oxygen glucose deprivation and reperfusion in PC12 cells. Authors succeeded to combine single cell amperometry with intracellular vesicle impact electrochemical cytometry (method was introduced by Ewing's group) to measure exocytosis and vesicular storage of catecholamine content. The paper is well written and from general view may be regarded for publication in QRB Discovery. Authors must address several questions before the final recommendation would be issued by this reviewer.

On Fig.2 very high basal calcium level is observed after combination of zinc and OGDR (or OGR? as indicated on purple line). How can explain this effect?

Why did cells with zinc and OGDR treatments become insensitive to depolarization with 100 mM KCl? Did exocytosis in this condition become calcium independent? If Zn is Ca-channel blocker, why was the effect of Zn moderate in Zn only experiments (blue line)?

As indicated in ref. 21 an increased expression of mRNA for oxygen sensitive K+ channels has been observed after 18 hours of chronic hypoxia. In present study OGD lasted 4 hours only. Is it fair to use data from ref. 21 to explain the observed OGDR-induced increased level of Ca influx?

The figure legend to fig.2 must be corrected. The changes in intracellular calcium levels were induced by depolarization with KCl but not by exocytosis. Addition of KCl needs to be indicated.

---

## [Reviewer Report]

*Comments to Author*: The manuscript deals with the important problem of the effect of zinc on the release of neurotransmitters. Using an elegant experimental approach, the authors found that OGDR increases the amount of catecholamine released during exocytosis and vesicular storage of catecholamine, resulting in a higher proportion of release. However, zinc treatment weakens the increased amount of catecholamine released and the vesicle content caused by OGDR, so the proportion of release remains unchanged. Zinc accumulation during exocytosis and accumulation in vesicles during ischemia may also help to understand ischemia-related changes in synaptic plasticity and memory disorders. The results sound bold, and the expected impact looks significant.

Comments.

1. The experimental protocol is not well described. It is mentioned that exocytosis was caused by a 100 mm solution of K+. However, the duration of this stimulation is not specified, and the stimulation is not shown in Figures 1, 2 and 4.

2. Changes in the frequency of exocytic events are mentioned in the results section, but this is not confirmed by any data, since only the total number of events is analyzed.

3. All data is shown as an average of +/- SEM. Since SEM directly depends on the number of measurements, this value is not informative unless the number of experiments is specified. Moreover, the comparison of SEM values is not representative for an unequal number of measurements. This presentation of data is inappropriate. The exact number of experiments should be specified for all values (including the values in the figures). Alternatively, the data can be represented as an average of +/- SD.

4. Figure 2. The legend is incorrect. This actually shows a comparison of Ca2+ levels before, during and after stimulation. Again, it is necessary to show stimulation.

5. Figure 2. For OGD+zinc conditions, the calcium level is very high before stimulation and does not change much. This unusual behavior must be explained

6. Line145. Probably, incorrect reference to figure 2D.

---

## [Reviewer Report]

*Comments to Author*: Reviewer #1: The paper of Ying Wang et al.is devoted to clarification of the effect of zinc on exocytosis

and vesicular storage of catecholamine observed after ischemia modeled by oxygen glucose deprivation and reperfusion in PC12 cells. Authors succeeded to combine single cell amperometry with intracellular vesicle impact electrochemical cytometry (method was introduced by Ewing's group) to measure exocytosis and vesicular storage of catecholamine content. The paper is well written and from general view may be regarded for publication in QRB Discovery. Authors must address several questions before the final recommendation would be issued by this reviewer.

On Fig.2 very high basal calcium level is observed after combination of zinc and OGDR (or OGR? as indicated on purple line). How can explain this effect?

Why did cells with zinc and OGDR treatments become insensitive to depolarization with 100 mM KCl? Did exocytosis in this condition become calcium independent? If Zn is Ca-channel blocker, why was the effect of Zn moderate in Zn only experiments (blue line)?

As indicated in ref. 21 an increased expression of mRNA for oxygen sensitive K+ channels has been observed after 18 hours of chronic hypoxia. In present study OGD lasted 4 hours only. Is it fair to use data from ref. 21 to explain the observed OGDR-induced increased level of Ca influx?

The figure legend to fig.2 must be corrected. The changes in intracellular calcium levels were induced by depolarization with KCl but not by exocytosis. Addition of KCl needs to be indicated.

Reviewer #2: The manuscript deals with the important problem of the effect of zinc on the release of neurotransmitters. Using an elegant experimental approach, the authors found that OGDR increases the amount of catecholamine released during exocytosis and vesicular storage of catecholamine, resulting in a higher proportion of release. However, zinc treatment weakens the increased amount of catecholamine released and the vesicle content caused by OGDR, so the proportion of release remains unchanged. Zinc accumulation during exocytosis and accumulation in vesicles during ischemia may also help to understand ischemia-related changes in synaptic plasticity and memory disorders. The results sound bold, and the expected impact looks significant.

Comments.

1. The experimental protocol is not well described. It is mentioned that exocytosis was caused by a 100 mm solution of K+. However, the duration of this stimulation is not specified, and the stimulation is not shown in Figures 1, 2 and 4.

2. Changes in the frequency of exocytic events are mentioned in the results section, but this is not confirmed by any data, since only the total number of events is analyzed.

3. All data is shown as an average of +/- SEM. Since SEM directly depends on the number of measurements, this value is not informative unless the number of experiments is specified. Moreover, the comparison of SEM values is not representative for an unequal number of measurements. This presentation of data is inappropriate. The exact number of experiments should be specified for all values (including the values in the figures). Alternatively, the data can be represented as an average of +/- SD.

4. Figure 2. The legend is incorrect. This actually shows a comparison of Ca2+ levels before, during and after stimulation. Again, it is necessary to show stimulation.

5. Figure 2. For OGD+zinc conditions, the calcium level is very high before stimulation and does not change much. This unusual behavior must be explained

6. Line145. Probably, incorrect reference to figure 2D.